# Assessing Prognosis: Factors Influencing Outcomes in Hospitalized Lung Cancer

Jesús Peña-López [1,*], Laura Gutiérrez-Sainz [1], Diego Jiménez-Bou [1], Icíar Ruíz-Gutiérrez [1],
Carmen Navas-Jiménez [1], Jorge Ignacio Alonso-Eiras [1], Álvaro García-Zamarriego [1], Darío Sánchez-Cabrero [1],
Leticia Ruíz-Giménez [1], Ana Pertejo-Fernández [1], Julia Villamayor-Sánchez [1], Patricia Cruz-Castellanos [2],
Oliver Higuera-Gómez [1] and Javier de Castro [1]

[1] Department of Medical Oncology, Hospital Universitario La Paz, 28046 Madrid, Spain;
lauragutierrezsainz@gmail.com (L.G.-S.); diegojbou@gmail.com (D.J.-B.); iruiz.11@alumni.unav.es (I.R.-G.);
carmennavasjimenez@gmail.com (C.N.-J.); jorgealonso97eiras@gmail.com (J.I.A.-E.);
soyalvarogz1997@gmail.com (Á.G.-Z.); dario.sc88@gmail.com (D.S.-C.);
letiruizgimenez@hotmail.com (L.R.-G.); pertejo.ana@gmail.com (A.P.-F.); villamayor.julia@gmail.com (J.V.-S.);
oliverhiguera@gmail.com (O.H.-G.); javierdecastro5@gmail.com (J.d.C.)
[2] Department of Medical Oncology, Hospital General Universitario de Ciudad Real, 13005 Ciudad Real, Spain;
patriciacruzcastellanos@gmail.com
[*] Correspondence: jpenal@salud.madrid.org

**Simple Summary:** Patients with lung cancer have a high admission rate, a situation that generally implies a poor prognosis in oncological patients. We analyzed the characteristics and potential prognostic factors of 158 patients with lung cancer who required admission. Overall survival since hospital admission was low, with a median of 3.3 months. In our study population, the prognostic factors independently associated with worse survival were admission due to tumor-related causes (aHR 1.81, 95%CI 1.05–3.11) and ECOG $\geq$ 2 ($p = 0.041$, aHR 1.80, 95%CI 1.03–3.16). Therefore, admission in patients with lung cancer is a poor prognostic event, especially if it is associated with tumor-related causes or a decline in performance status.

**Abstract:** Background: Patients with lung cancer experience higher rates of hospitalization due to their elevated mortality and associated comorbidities. Hospital admissions among oncology patients often indicate organ vulnerability and are linked to poor prognosis. This study aimed to assess the characteristics and potential prognostic factors of hospitalized lung cancer patients. Methods: We retrospectively analyzed 646 patients admitted from June 2021 to May 2022 to the Medical Oncology Service at La Paz University Hospital (Madrid, Spain). Results: During this period, 158 patients admitted had lung cancer (24.5%). The median overall survival since admission (mOSSA) was 3.3 months (95%CI: 1.86–7.74). In the univariate analysis, poorer mOSSA was associated with admission for tumor-related causes (1.33 vs. 7.30 months, $p < 0.001$), ECOG $\geq$ 2 (2.43 vs. 8.50 months, $p < 0.001$), NLR $\geq$ 6 (1.87 vs. 7.40 months), PNI $\leq$ 40 (1.67 vs. 4.97 months), and LDH $\geq$ 210 (2.27 vs. 7.87 months, $p = 0.044$). In the multivariate analysis, independent prognostic factors included admission for tumor-related causes ($p = 0.032$, aHR 1.81, 95%CI: 1.05–3.11) and ECOG $\geq$ 2 ($p = 0.041$, aHR 1.80, 95%CI: 1.03–3.16). Conclusions: Hospital admission for lung cancer is a poor prognostic event, particularly when associated with tumor-related causes or a decline in performance status.

**Keywords:** lung cancer; admission; hospitalization; prognosis; survival; immuno-nutritional; irAEs





## 1. Introduction

Lung cancer is a global health issue. It was the most frequently diagnosed cancer in 2022, accounting for nearly 2.5 million new cases or approximately one in eight cancer diagnoses worldwide (12.4% of all cancers). It is also the leading cause of cancer-related

mortality, responsible for an estimated 1.8 million deaths in 2022 (18.7% of all deaths from cancer) [1].

Patients with lung cancer often exhibit a high rate of frailty, which is associated with a poorer prognosis [2]. Lung cancer is a particularly aggressive disease, with approximately 44% of patients being diagnosed at a metastatic stage [3]. The 5-year survival rate in patients with late-stage disease is often less than 5% [4]. Compounding this issue, many lung cancer patients face significant health challenges due to smoking-related comorbidities. It is reported that about 50% of lung cancer patients have three or more comorbidities, such as chronic obstructive pulmonary disease (COPD) or cardiovascular disease [5]. Additionally, there is a high prevalence of elderly patients (nearly 47% of lung cancer patients are over 70 years of age) [6]. These factors often intersect with another critical concern: disease-related malnutrition. Upon admission, up to 43% of lung cancer patients may be at risk of malnutrition [7]. Malnutrition implies an increased risk of anticancer treatment-related toxicity, lower quality of life (QoL) and higher mortality [8].

Together, these factors help to explain why lung cancer patients are among the cancer subgroups with the highest risk of unplanned hospital admission [9]. This risk of admission can also indicate a "sentinel event", signaling greater organ dysfunction and a worse prognosis, often correlating with an overall survival time of less than six months [10]. From an economic perspective, the significance of unplanned hospital admissions is also noteworthy; in 2009, 56% of cancer-related healthcare costs in the European Union were attributed to inpatient care [11]. Thus, reducing unplanned acute care is a crucial objective in optimizing cancer management, with one strategy being the identification of high-risk patients [12].

Several methods are available for the stratification of prognosis in oncology [13]. The Eastern Cooperative Oncology Group (ECOG) Performance Status (PS) scale measures how the disease impacts a patient's daily living abilities (Table 1) [14]. Additionally, immuno-nutritional or inflammatory scores, easily derived from blood tests, aim to reflect the chronic inflammatory state and malnutrition induced by cancer:

**Table 1.** Eastern Cooperative Oncology Group (ECOG) Performance Status (PS) scale [14].

| Grade | Performance Status |
|---|---|
| 0 | Fully active, able to maintain pre-disease performance without restriction |
| 1 | Restricted in physically strenuous activity but ambulatory and able to carry out work of a light or sedentary nature |
| 2 | Ambulatory and capable of all self-care but unable to carry out any work activities; active for more than 50% of waking hours |
| 3 | Capable of only limited self-care; confined to bed or chair more than 50% of waking hours |
| 4 | Completely disabled; cannot carry out any self-care; totally confined to bed or chair |
| 5 | Dead |

- Neutrophil/lymphocyte (NLR) = neutrophils/lymphocytes [15];
- Prognostic nutritional index (PNI) = albumin (g/dL) $\times$ 10 + lymphocytes/$\mu$L $\times$ 0.005 [16];
- Glasgow Prognostic Score Modified (mGPS): 1 point if C-reactive protein (PCR) (mg/L) > 10 $\rightarrow$ 2 points if CRP (mg/L) > 10 and albumin (g/dL) < 3.5 [17];
- CONtrolling NUTritional status (CONUT) (Table 2) [18].

**Table 2.** CONtrolling NUTritional status (CONUT) score [18]. The sum of the score of each parameter (which depends on its value) determines the degree of undernutrition.

| Parameter | Normal | Light Undernutrition | Moderate Undernutrition | Severe Undernutrition |
|---|---|---|---|---|
| Albumin (g/dL) | 3.5–4.5 (0) | 3.0–3.49 (2) | 2.5–2.9 (4) | <2.5 (6) |
| Lymphocytes/μL | >1600 (0) | 1200–1599 (1) | 800–1199 (2) | <800 (3) |
| Cholesterol (mg/dL) | >180 (0) | 140–180 (1) | 100–139 (2) | <100 (3) |
| Screening Total Score | 0–1 | 2–4 | 5–8 | 9–12 |

In recent years, there has been a paradigm shift in the treatment of advanced lung cancer. Targeted therapies have become the standard of care for tumors with driver mutations [19]. In lung tumors without driver mutations, immune checkpoint inhibitors (ICIs) with or without chemotherapy are the mainstay of treatment [20]. While ICIs are generally well tolerated, they can lead to immune-related adverse events (irAEs), which are often difficult to predict, diagnose, or treat [21]. The current guidelines lack robust, trial-based recommendations, leading to uncertainty about whether the benefits observed in clinical trials extend to hospitalized populations and how the toxicity of these drugs impacts hospital admissions.

The objective of this study is to describe the characteristics of patients hospitalized with lung cancer and determine which ones are associated with the worst prognosis.

## 2. Materials and Methods

### 2.1. Patients

This descriptive retrospective study was conducted between June 2021 and May 2022 at the Medical Oncology Service of La Paz University Hospital in Madrid, Spain. The study included adult patients (>18 years old) who were admitted during this period. Our analysis specifically focused on those within this cohort who had lung cancer. These patients were followed up until February 2023. Demographic, clinical, pathological, and analytical data were collected from their medical records. The information extracted at the time of admission included sex, age, histology, TNM stage (8th edition AJCC), date of cancer diagnosis, previous oncospecific treatments, reason for hospital admission, ECOG PS at admission, results from the first blood test at admission, and the length of the hospital stay. During the follow-up period, we also recorded subsequent admissions, emergency department visits, the survival status (alive or deceased), and the date of death or last contact.

The "tumor-related" cause of admission refers to issues arising from the tumor, including cachexia, atelectasis, lymphangitis, pleural effusion, other non-lung organ failure due to metastasis (e.g., neurological clinic, hepatic failure), paraneoplastic syndrome, or the need to initiate oncological treatment during hospitalization.

The study was conducted in accordance with the Declaration of Helsinki and approved by the Ethics Committee of Hospital Universitario La Paz (protocol code PI-6135).

### 2.2. Statistical Analysis

The prevalence of patient characteristics was analyzed using descriptive statistics. We used the chi-squared test for categorical variables and Student's *t*-test for continuous variables to compare the characteristics across the groups. Correlations between categorical variables were examined with Spearman's rho. To determine optimal cut-off points for continuous variables, we applied the Youden index. Survival from the time of admission was estimated using the Kaplan–Meier method. Univariate and multivariate analyses were conducted using a Cox regression model to calculate hazard ratios (HR), adjusted hazard ratios (aHR), and 95% confidence intervals (95% CI) for each prognostic factor. To evaluate the discriminatory ability of the model, we calculated the area under the curve (AUC) for

the receiver operating characteristic (ROC) curves. All statistical analyses were performed using IBM SPSS Statistics for Windows, Version 19.0 (IBM Corporation, Armonk, NY, USA).

## 3. Results

### 3.1. Characteristics of Patients

Out of the 646 patients admitted during this period, 158 (24.5%) had lung cancer. Their baseline characteristics and the primary causes of admission are summarized in Table 3. The median follow-up period from the time of admission was 14.9 months (95%CI: 12.0–17.8).

**Table 3.** Baseline characteristics. "*" indicates the subgroups within the group immediately above.

| Baseline Characteristics | N = 158 |
|---|---|
| Age (years)—Median (range) | 68 (27–89) |
| Sex—*n* (%) | |
| - Male | 105 (66.5%) |
| - Female | 53 (33.5%) |
| Stage—*n* (%) | |
| - I | 2 (1.3%) |
| - II | 4 (2.5%) |
| - III | 18 (11.4%) |
| - IV | 134 (84.8%) |
| Histology—*n* (%) | |
| - Non-small cell | 127 (80.4%) |
| * Adenocarcinoma | 76 (48.1%) |
| * Squamous cell | 35 (22.2%) |
| * Undifferentiated | 14 (8.9%) |
| * Giant cell | 2 (1.3%) |
| - Small cell | 29 (18.4%) |
| - Other (salivary gland, mesothelioma) | 2 (1.2%) |
| Driver mutation—*n* (%) | 28 (17.7%) |
| - EFGR | 17 (10.8%) |
| - ALK | 4 (2.5%) |
| - ROS | 1 (0.6%) |
| - KRAS | 5 (3.2%) |
| - BRAF | 1 (0.6%) |
| Previous surgery—*n* (%) | 13 (8.2%) |
| Previous radiotherapy—*n* (%) | 56 (35.4%) |
| Systemic treatment at admission—*n* (%) | 133 (84.1%) |
| - (Neo-)adjuvant | 9 (5.7%) |
| - First line | 74 (46.8%) |
| - Second line | 25 (15.8%) |
| - Third line | 17 (10.8%) |
| - Fourth line | 8 (5.1%) |
| Has received ICI—*n* (%) | 92 (58.2%) |
| - At the moment of admission | 61 (38.6%) |
| - Previously | 31 (19.6%) |
| ECOG at admission—*n* (%) | |
| - 0 | 8 (5.1%) |
| - 1 | 44 (27.8%) |
| - 2 | 52 (32.9%) |
| - 3 | 43 (27.2%) |
| - 4 | 11 (7.0%) |
| Main cause of admission recorded—*n* (%) | |
| - Tumor-related | 68 (43%) |
| * Start of oncospecific treatment | * 6 (3.8%) |
| - Infection | 51 (32%) |
| - Immune-related adverse events | 12 (7.6%) |
| - Non-infectious chemotherapy-related | 5 (3.2%) |
| - Heart failure | 4 (2.5%) |
| Fever present on admission—*n* (%) | 50 (31.6%) |

The median overall survival (mOS) from diagnosis was 16.5 months (95%CI: 12.0–21.0), while the median overall survival since admission (mOSSA) was 3.3 months (95%CI: 1.86–7.74). The median length of hospital stay was 8 days (range: 0–81). The rate of readmission or emergency department visits was 60% (*n* = 97), with a median time of 33 days post-discharge (range: 1–508).

During hospitalization or follow-up, 121 patients (76.5%) died, with 64 (40.5%) dying within 30 days of admission and 27 (17.1%) passing away during their hospital stay. Eighteen patients received palliative care follow-up, and 25 required palliative sedation due to refractory symptoms, including dyspnea (*n* = 12), delirium (*n* = 10), pain (*n* = 2), and vital distress (*n* = 1).

### 3.2. Analytical Parameters and Immuno-Nutritional/Inflammatory Scores

The laboratory parameters and calculated scores are summarized in Table 4.

**Table 4.** Analytical parameters and immuno-nutritional/inflammatory scores.

| Variable | |
|---|---|
| Hb (g/dL)—median (range) | 12.1 (6.4–17.3) |
| NLR—median (range)<br>- NLR ≥ 6—*n* (%) | 12.4 (0.2–256.6)<br>90 (57%) |
| Albumin (g/dL)—median (range) | 7.6 (2.4–8.1) |
| Cholesterol (mg/dL)—median (range) | 159 (66–315) |
| CRP (mg/L)—median (range) | 114 (0.5–315) |
| LDH (UI/L)—median (range)<br>- LDH ≥ 210—*n* (%) | 410 (110–4124)<br>109 (69%) |
| PNI—median (range)<br>- PNI ≤ 40—*n* (%) | 42.3 (24.1–87.5)<br>51 (32%) |
| mGPS—*n* (%)<br>- 0 points<br>- 1 point<br>- 2 points | <br>10 (6.3%)<br>84 (53.2%)<br>43 (27.2%) |
| CONUT—*n* (%)<br>- Normal<br>- Light undernutrition<br>- Moderate undernutrition<br>- Severe undernutrition | <br>17 (10.7%)<br>73 (46.2%)<br>32 (20.3%)<br>8 (5.1%) |

### 3.3. ICIs in Hospitalized Patients

Ninety-two patients (58.2%) had received ICIs. At the time of admission, 63 patients (39.9%) were undergoing ICI treatment, with 16 (10.1%) receiving ICI monotherapy. Twenty-nine patients (18.3%) had previously received ICIs in earlier treatment lines. The best response to ICIs was a complete response (*n* = 3), partial response (*n* = 33), stable disease (*n* = 27), and progression of disease (*n* = 29).

There were no significant differences in sex, age, stage, cause of admission, ECOG, Hb, NLR, PNI, LDH, CONUT, or mGPS when stratifying the patients according to whether they were receiving immunotherapy at the time of admission.

In 12 patients (7.6%), the reason for admission was suspected irAEs. It was the third reason for admission after tumor-related events (43%) and infections (32.3%). The median age was 71 years, and 66.7% were male. In one patient, irAEs occurred during adjuvant/neoadjuvant therapy. The detected irAEs, in order of frequency, were pneumonitis (*n* = 7), gastritis (*n* = 1), hepatitis (*n* = 1), nephritis (*n* = 1), myocarditis–myositis–myastheniform (*n* = 1), and hypophysitis (*n* = 1). Possible irAEs affecting a second organ were recorded in five patients. The median number of immunotherapy cycles prior to

toxicity was nine (range 5.50–11.50). All irAEs were treated with corticosteroids, with the exception of endocrine toxicity. The median maximum daily dose was 100 mg prednisone (range 100–145). Three patients were corticosteroid-resistant, requiring bolus corticosteroids for one case of pneumonitis, infliximab for another case of pneumonitis, and a combination of bolus corticosteroids and plasmapheresis for myocarditis. Four patients died due to irAEs: pneumonitis ($n = 3$) and myocarditis ($n = 1$). ICI treatment was discontinued in all survivors, with irAE relapse occurring in three patients.

### 3.4. Prognostic Factors

The analysis of the ROC curves indicated significant discriminatory power for 30-day mortality based on several factors. The ECOG performance status demonstrated significance, with a *p*-value of 0.003 and an area under the curve (AUC) of 0.638 (95% CI: 0.55–0.73). Similarly, the neutrophil-to-lymphocyte ratio (NLR) was significant ($p = 0.035$, AUC 0.599, 95% CI: 0.51–0.69), as was lactate dehydrogenase (LDH), which showed a *p*-value of 0.002 and an AUC of 0.653 (95% CI: 0.56–0.75) (Figure 1). The optimal cut-off values derived from the ROC curves were established as follows: ECOG $\geq$ 2 (Youden index = 0.29), NLR $\geq$ 6 (Youden index = 0.20), PNI $\leq$ 40 (Youden index = 0.10), and LDH $\geq$ 210 (Youden index = 0.17).

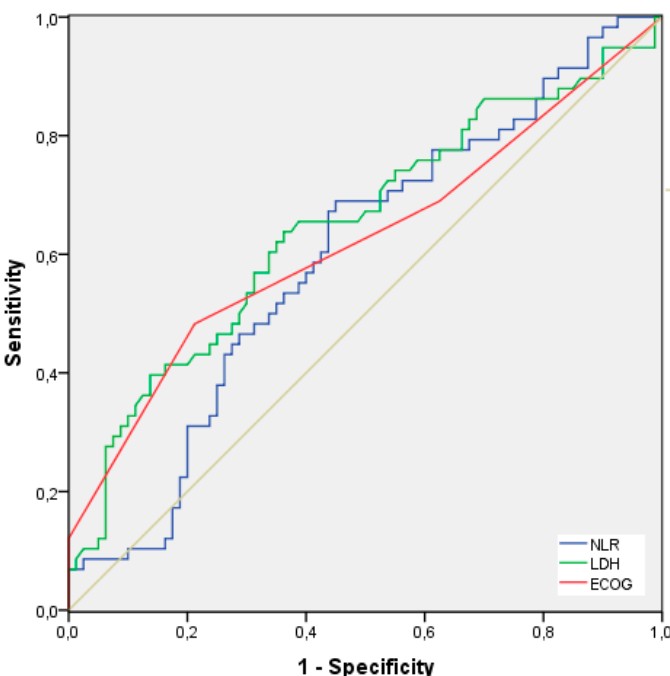

**Figure 1.** ROC curve to evaluate the discriminatory ability of 30-day mortality of NLR—blue ($p = 0.035$, AUC 0.599, 95%CI 0.51–0.69)—and LDH—green ($p = 0.002$, AUC 0.653, 95%CI 0.56–0.75).

In the univariate analysis, several factors were significantly associated with poorer mOSSA. Admission due to tumor-related causes was associated with a mOSSA of 1.33 months, compared to 7.30 months for other causes ($p < 0.001$, HR 2.19, 95%CI: 1.53–3.15). An ECOG score of $\geq$2 was linked to a mOSSA of 2.43 months versus 8.50 months ($p < 0.001$, HR 2.15, 95%CI: 1.42–3.25). Other factors included NLR $\geq$ 6 (1.87 months vs. 7.40 months, $p = 0.006$, HR 1.67, 95%CI: 1.16–2.42), PNI $\leq$ 40 (1.67 months vs. 4.97 months, $p = 0.044$, HR 1.49, 95%CI: 1.01–2.20), and LDH $\geq$ 210 (2.27 months vs. 7.87 months, $p = 0.044$, HR 1.68, 95%CI: 1.01–2.79) (Figure 2). No significant associations were found between survival and other factors, such as sex ($p = 0.84$), age ($p = 0.94$), stage IV disease ($p = 0.88$), the number of treatment lines ($p = 0.65$), the CONUT score ($p = 0.70$), or mGPS ($p = 0.07$). Among patients treated with immune checkpoint inhibitors (ICIs), the best response to treatment did not correlate with survival from the time of admission ($p = 0.70$).

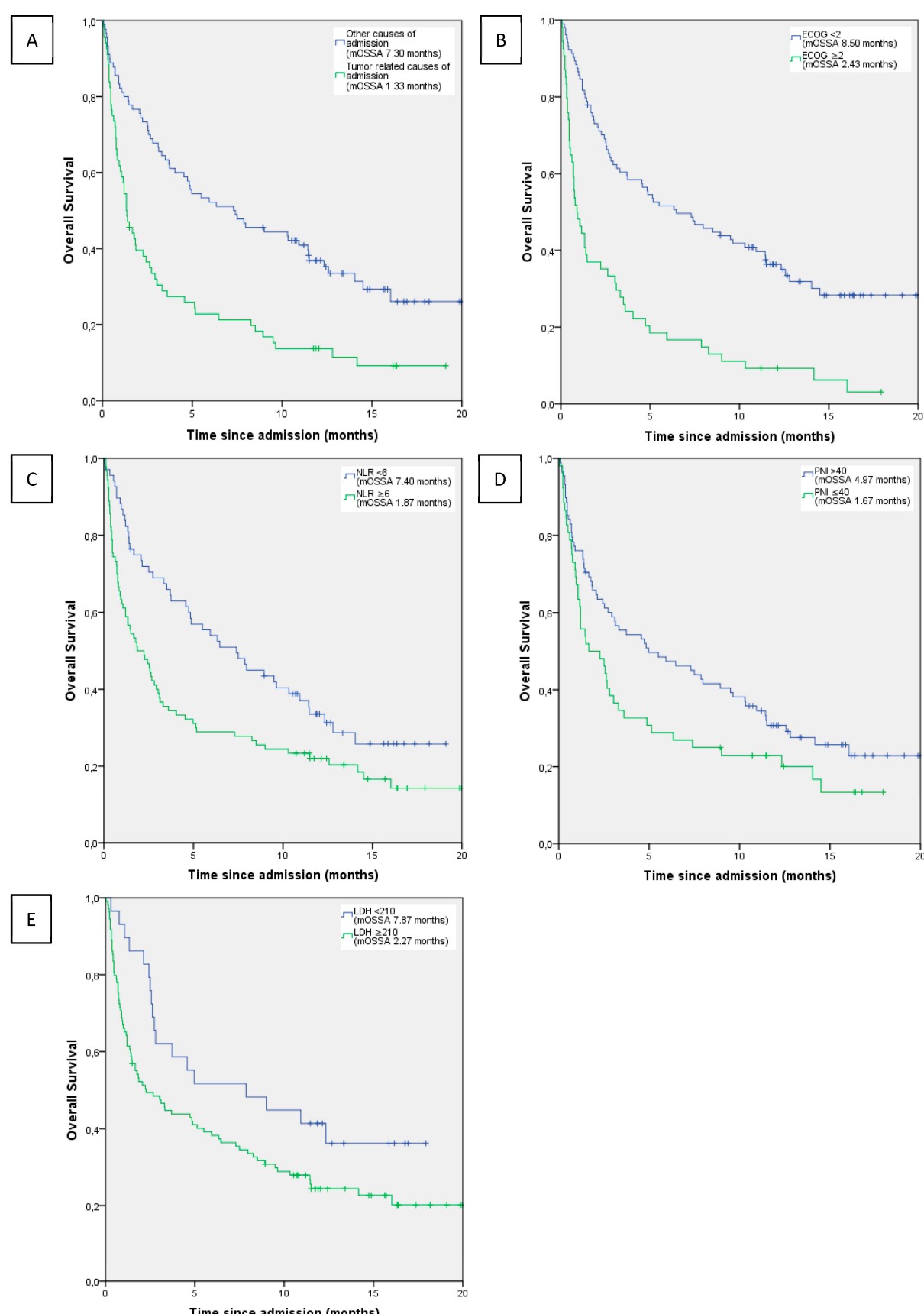

**Figure 2.** Survival curves for overall survival since admission stratified by cause of admission (**A**), ECOG (**B**), NLR (**C**), PNI (**D**), and LDH (**E**).

Notably, survival was not significantly associated with stage IV disease. Of the 158 patients analyzed, only 24 (15.2%) were classified as non-stage IV. When comparing these non-stage IV patients to those with stage IV disease, it was found that non-stage IV patients had a lower 30-day mortality rate (12.5% vs. 45.5%, $p = 0.004$) and lower mean LDH levels (306 UI/L vs. 421 UI/L, $p = 0.37$).

In the multivariate analysis, the independent prognostic factors identified were admission due to tumor-related causes ($p = 0.032$, aHR 1.81, 95%CI: 1.05–3.11) and an ECOG score of $\geq 2$ ($p = 0.041$, aHR 1.80, 95%CI: 1.03–3.16).

## 4. Discussion

Hospitalization in oncology patients often serves as a sentinel event indicating a poor prognosis. Understanding the underlying reasons and mechanisms behind cancer-related deaths is essential in developing effective strategies to extend life while maintaining quality of life. To achieve this, improving data recording and reporting, alongside conducting detailed observational studies, is crucial for the detection of biological signals [22]. Understanding prognosis can guide oncological treatment adjustments. While patients with functional impairments generally have worse prognosis, they may still benefit from oncospecific treatments. The IPSOS study demonstrated the efficacy of atezolizumab in lung cancer patients with ECOG 2 [23]. Moreover, adjusting chemotherapy regimens for frail, elderly patients with metastatic lung cancer can yield similar efficacy with reduced toxicity [24]. This has also been demonstrated for other types of cancer, like colorectal or gastroesophageal cancer [25–27].

Our population-based sample confirms that lung cancer is the subtype most associated with hospital admissions, accounting for 24.5% of the total admissions. This aligns with the findings from other studies, reinforcing the notion that lung cancer patients face a heightened risk of hospitalization due to the severity of their condition. Notably, our study highlights the poor prognosis linked to these admissions, evidenced by a median survival from admission of 3.3 months, an in-hospital mortality rate of 17.1%, and a 30-day mortality rate of 40.5%. For comparison, an Italian study involving 251 lung cancer patients reported a 30-day mortality rate of 18.3% [28]. This difference may arise from the primary reasons for admission; in their study, 58.2% of admissions were for oncospecific treatment, whereas, in our cohort, 43% were due to tumor-related complications, with only 3.8% starting oncospecific treatment upon admission. Despite this, the median overall survival since admission in the Italian study was slightly higher at 4.7 months. Our 30-day mortality rate was similar to a UK study, which reported a rate of 50% [29]. Additionally, our in-hospital mortality rate was similar to that observed in a Dutch series of 1322 patients from 2013 to 2015, which reported an 18% mortality rate [30].

In terms of symptoms, fever was present in 31.6% of our patients, while a Greek study found fever to occur in 45% of cases, highlighting variability across populations [31]. Dyspnea was the leading cause for the requirement of palliative sedation in our study, further emphasizing the need for tailored palliative care interventions. Notably, two-thirds of the patients who died during hospitalization were already receiving palliative care follow-up, supporting evidence that such care can positively impact both survival and quality of life [32].

When evaluating the prognostic factors through ROC curve analysis, NLR, LDH, and ECOG were found to have discriminatory power in predicting 30-day mortality. Significant differences in median overall survival since admission (mOSSA) were observed when stratifying by clinical factors (admission due to tumor-related causes and ECOG $\geq 2$) and analytical parameters (NLR $\geq 6$, PNI $\leq 40$, and LDH $\geq 210$). However, only the clinical factors remained as independent prognostic indicators in the multivariate analysis. This suggests that chronic inflammation and malnutrition associated with oncologic disease are reflected in both clinical and analytical parameters, with clinical factors proving more robust predictors of survival. This finding is consistent with the results from a multicenter

cohort validation study, which showed that no prognostic algorithms for palliative care were superior to the clinical prediction of survival (CPS) [33].

Interestingly, stage IV disease did not correlate with survival in our study. However, when analyzing the stage IV vs. non-stage IV subgroups, it was seen that the proportion of 30-day mortality was higher in the stage IV subgroup. We believe that this seemingly contradictory outcome is possibly due to selection bias, as stage IV patients are more likely to be hospitalized, leading to their overrepresentation. This is reasonable as stage IV disease constitutes a significant proportion of new lung cancer diagnoses, and many locoregional cases progress to this stage [34,35]. The low percentage of non-IV stages in our sample may have led to insufficient statistical power and made it difficult to detect the true prognostic role of the oncological stage in admitted patients.

Numerous studies have attempted to determine potential prognostic factors in hospitalized cancer patients (Table 5).

**Table 5.** Summary of reported retrospective analyses of prognostic factors in hospitalized cancer patients. PPI = Palliative Prognostic Index. PaP Score = Palliative Prognostic Score. KPS = Karnofsky Performance Status. CPS = Clinical Prediction of Survival. BRASS = Blaylock Risk Assessment Screening Score.

| Article | Population |
| --- | --- |
| Morita et al., 1999 [36] | PPI: ECOG, oral intake, edema, dyspnea at rest, and delirium |
| Maltoni et al., 1999 [37] | PaP Score: dyspnea, anorexia, KPS, CPS, and leukocytes |
| Barbot et al., 2008 [38] | KPS, number of metastatic sites, albumin, and LDH |
| Feliu et al., 2011 [39] | ECOG, LDH, albumin, lymphocytes, and time from initial diagnosis to diagnosis of terminal disease |
| Leonetti et al., 2023 [28] | BRASS |
| Mirallas et al., 2023 [40] | ECOG, LDH, albumin, neutrophils, lung metastases, and treatment response at admission |
| Our series | Admission for tumor-related causes and ECOG |

Our study contributes to this body of literature by emphasizing the role of immune checkpoint inhibitors (ICIs) in hospitalized lung cancer patients. Despite over half of our patients having received ICIs, we found no significant differences in outcomes compared to those who had not. This suggests that the survival outcomes for hospitalized lung cancer patients have not markedly improved with the introduction of ICIs. Notably, suspicion of immune-related adverse events (irAEs) was the third most common cause for admission, with one-third of these patients dying and ICI treatment discontinued in all survivors.

In conclusion, our study underscores the unique clinical complexities of hospitalized lung cancer patients, who differ significantly from those typically included in clinical trials. This highlights the need to integrate real-world data into hospitalization protocols to better inform decisions regarding the treatment intensity and appropriateness. Our findings reaffirm the importance of the ECOG and the cause of admission as prognostic tools in this vulnerable population, emphasizing that these factors should be systematically incorporated into clinical protocols. As previously discussed, frailer patients may benefit from dose modifications in oncospecific therapy, and hospitalization may be a time to reassess frailty (or evaluate it if not previously performed). For the most fragile patients, these strategies may include close collaboration with palliative care teams and ensuring robust social and family support systems at home. Additionally, these protocols could be further refined by incorporating biomarkers such as NLR, PNI, and LDH. Hospitalized patients without clinical risk data (admission due to tumor-related causes and ECOG $\geq$ 2) but with analytical risk data (NLR $\geq$ 6, PNI $\leq$ 40, and LDH $\geq$ 210) may constitute an intermediate population that would benefit from close follow-up. In Figure 3, we propose an action algorithm that reflects this work philosophy.

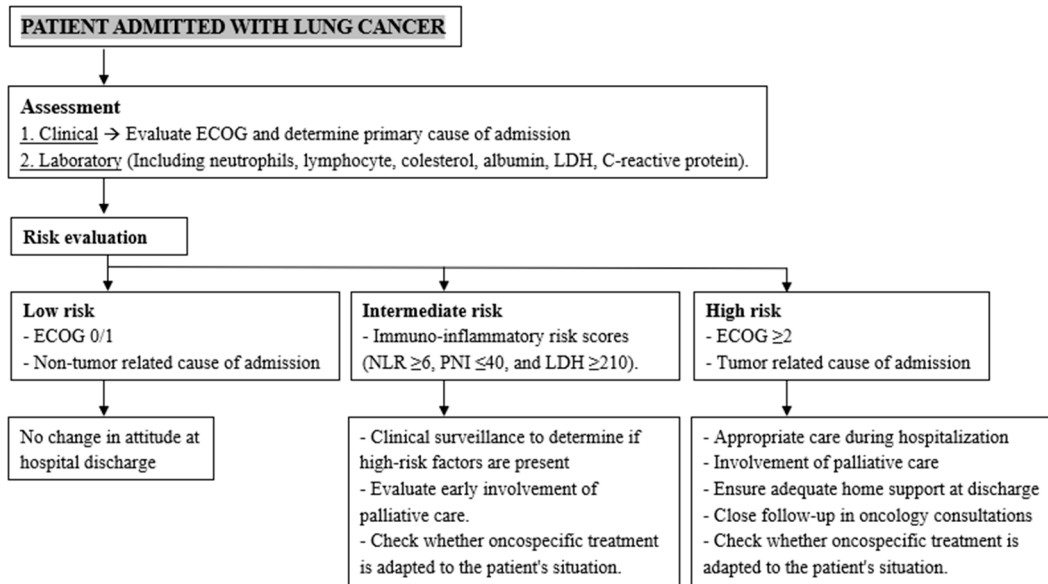

**Figure 3.** Flow algorithm for hospitalized lung cancer patients' management.

The limitations of our study include its single-center retrospective design and small sample size. We also did not perform a specific analysis regarding the impact of the smoking status and comorbidities (e.g., COPD) on the outcomes. To address these limitations, future research should consider conducting multicenter prospective studies, which would allow for a more diverse patient population and reduce the potential for selection and information biases. A prospective design would also enable the collection of more detailed and standardized data, further validating the prognostic factors identified in this study.

Another aspect to be assessed is that, while weight can indeed be an important clinical parameter, we chose to focus on inflammatory scores in our study due to several key considerations. They are objective and easily obtainable metrics, which is particularly advantageous in retrospective studies as they minimize the potential biases associated with the incomplete or inconsistent documentation of weight changes. Moreover, they predict survival independently of the tumor or disease stage in various cancers and are associated with cancer-related cachexia [41]. The routine clinical monitoring of systemic inflammation markers could help to identify patients at risk of developing cachexia, even those without weight loss and with a good performance status [42].

Another potential criticism of this algorithm is that the basis for improving patient quality is outpatient follow-up and that post-admission management may occur too late. Nevertheless, inpatient care is likely to be an inevitable cancer trajectory [43]. We believe that the optimization of supportive procedures during and after admission should be a key goal of medical oncology.

## 5. Conclusions

Among all cancer types, patients with lung cancer have the highest frequency of hospital admissions, reflecting a concerning prognosis. These patients face a high rate of unplanned return visits, limited survival following admission, and a considerable risk of in-hospital mortality. Our results suggest that admissions due to tumor-related causes and an ECOG ≥ 2 are independently associated with worse survival outcomes. These results underscore the need for early coordination with palliative care services to provide comprehensive care and support, aiming to improve the quality of life and potentially extend the survival time in this high-risk population.

**Author Contributions:** Conceptualization, J.P.-L. and L.G.-S.; methodology, J.P.-L. and L.G.-S.; formal analysis, J.P.-L., I.R.-G. and D.J.-B.; investigation, J.P.-L., C.N.-J., J.I.A.-E., Á.G.-Z., D.S.-C., L.R.-G., A.P.-F., J.V.-S., P.C.-C., O.H.-G. and J.d.C.; data curation, J.P.-L.; writing—original draft preparation, J.P.-L. and I.R.-G.; writing—review and editing, J.P.-L. and L.G.-S.; visualization, I.R.-G. and D.J.-B.; supervision, D.S.-C., L.R.-G., A.P.-F., J.V.-S., P.C.-C., O.H.-G. and J.d.C. All authors have read and agreed to the published version of the manuscript.

**Funding:** This research received no external funding.

**Institutional Review Board Statement:** The study was conducted in accordance with the Declaration of Helsinki and approved by the Ethics Committee of Hospital Universitario La Paz (protocol code PI-6135).

**Informed Consent Statement:** The local ethical committee approved the use of anonymized historical samples and data in this study and waived the requirement for informed consent from the patients.

**Data Availability Statement:** De-identified individual data might be made available following publication by reasonable request to the corresponding author.

**Conflicts of Interest:** The authors declare no conflicts of interest.

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
