# Peer review of "Assessing Prognosis: Factors Influencing Outcomes in Hospitalized Lung Cancer"

_onco, doi:10.3390/onco4040032_

Round 1

Reviewer 1 Report (Previous Reviewer 2)

Comments and Suggestions for Authors

Congratulations to you on your novel thought related to the caring of those unplanned hospital admission of lung cancer patients and the detailed analysis seeking out factors of survival and the way of improving. The following are my comments.

(1) In the text : Survival was not significantly associated with stage 4 disease. Comparing non-stage 4 with stage 4 disease, those non-stage 4 had a lower 30-day mortality(p=0.004). Do you think the two sentences above are contradictory ?

(2) Old frail patients with metastasis, adjusting the dosage or number of chemotherapy drugs can benefit. I think this is a treatment normality.

(3) In my opinion, improvement of survival and quality of life comes from the regular follow-up of patients in the out patient clinic and not improving the care after the unplanned hospital admission. Follow-up with blood tests including CBC, liver and renal functions, tumor markers, body weight, CAT scan, brain MRI and bone scan in 6-month interval. Hospitalization for further examinations in case of abnormal findings. Treat after unplanned admission will be to late.

(4) Although nutrition was mentioned, actually, body weight measuring is a very sensitive factor for monitoring of the patients' cancer status.

(5) I suggest the authors should implement the new strategy and compare the results with that of the previous existing one , to see if there is improvement in survival. The medical expenditure should also be contemplated for comparison.

Author Response

Comments 1: Congratulations to you on your novel thought related to the caring of those unplanned hospital admission of lung cancer patients and the detailed analysis seeking out factors of survival and the way of improving. The following are my comments.

Response 1: Thank you very much for the effort to correct our work. Your contributions allow us to improve.

Comments 2: (1) In the text : Survival was not significantly associated with stage 4 disease. Comparing non-stage 4 with stage 4 disease, those non-stage 4 had a lower 30-day mortality(p=0.004). Do you think the two sentences above are contradictory ?

Response 2: Thank you for pointing this out. We agree with this comment. Stage IV status itself did not have a significant impact on overall survival in the multivariate analysis. However, when analyzing the stage IV vs. non-stage IV subgroups, it was seen that the proportion of 30-day mortality was higher in the stage IV subgroup. Stage IV patients constituted the majority of our population (84.8%). It may be that the size of the non-stage IV subpopulation was so small that we did not have sufficient statistical power to detect differences. We reflect these problems and the limitation they pose in the lines 262-267 of discussion.

Comments 3: (2) Old frail patients with metastasis, adjusting the dosage or number of chemotherapy drugs can benefit. I think this is a treatment normality.

Response 3: Thank you for your observation. We agree that dose adjustment or modification in chemotherapy regimens is a necessary practice for frail, older patients with metastatic disease, aligning with standard treatment protocols. We aim to approach hospitalization in the lung cancer patient as an opportunity to assess the need to adjust treatment. Especially if there are clinical or analytical data of risk. We have corrected some aspects of the algorithm to reflect this (Line 297).

Comments 4: (3) In my opinion, improvement of survival and quality of life comes from the regular follow-up of patients in the out patient clinic and not improving the care after the unplanned hospital admission. Follow-up with blood tests including CBC, liver and renal functions, tumor markers, body weight, CAT scan, brain MRI and bone scan in 6-month interval. Hospitalization for further examinations in case of abnormal findings. Treat after unplanned admission will be to late.

Response 4: Thank you for your valuable insight. We agree that regular, proactive follow-up is essential to improving survival and quality of life for lung cancer patients by potentially preventing urgent hospitalizations through early detection of complications. Unfortunately, as observed in our study, lung cancer patients are the oncology population with the highest volume of hospitalization. This also implies a poor prognostic situation. Hospitalization may represent a sentinel event to rethink continuous outpatient care. By identifying the factors associated with these admissions, we aim to enhance our understanding of at-risk patient profiles, which could inform more tailored follow-up strategies in the outpatient setting. This approach may ultimately contribute to optimizing long-term care and support timely intervention before urgent admission becomes necessary.

Comments 5: (4) Although nutrition was mentioned, actually, body weight measuring is a very sensitive factor for monitoring of the patients' cancer status.

Response 5: Thank you for highlighting the importance of body weight monitoring as a sensitive indicator of cancer status. We acknowledge that refractory cachexia is a poor prognostic situation. The chronic inflammation that is the basis of this process is what the immunoinflammatory scores are intended to reflect. These scores are indicated in the ESMO guidelines as prognostic factors in advanced cancer (doi: 10.1016/j.esmoop.2023.101195). On the other hand, the diagnosis of some elements of cachexia (especially sarcopenia) can be complex and immunoinflammatory scores can be a simple measure to try to reflect chronic inflammation. In future considerations, we could include specific recommendations for monitoring body weight in our proposed outpatient follow-up care, as this could provide valuable insights into disease progression and help guide timely interventions.

Comments 6: (5) I suggest the authors should implement the new strategy and compare the results with that of the previous existing one , to see if there is improvement in survival. The medical expenditure should also be contemplated for comparison.

Response 6: Thank you for your valuable suggestion. Our objective with this work is to create a rationale from which to organize a prospective project to try to improve the continuum of care of patients on admission.

Reviewer 2 Report (New Reviewer)

Comments and Suggestions for Authors

The authors present a retrospective study of a population with lung cancer patients. The aim of the study is to describe the characteristics determining the cancer prognosis. They considered biological and clinical factors in univariate and multivariate analyses.

The topic is of interest, however there are many criticisms:

Title, line 2-3_ The title of the manuscript should be catchier and better present the purpose of the study inviting the reader to the topic.

Simple summary_ Line 15_ 'In our model', should be replaced with 'In our study population'

Abstract_ The sentence 'In the multivariate analysis only were independent prognostic factors admission due to tumor-related causes () and ECOG2.'  -Line 29-31_ should be reviewed.

Introduction_ This section contains useful information to better understand the topic, however the authors could present in a less schematic way (Ref. line 45-55 and 69-74. Furthermore, the authors should indicate the meaning of the ‘*’ in Table 3.

Results_ This section could be improved. For the reader the text is not easy to understand, because it is not very fluent in the terms and data presented. For this reason, they should be reorganized, also to better introduce the algorithm proposed for the management of hospitalized patients with lung cancer.

Discussion_ This section probably already revised should be further improved. Data from other studies should be better presented in parallel with the results of the retrospective analysis of the considered population. The flow algorithm presented in figure 3 should be better described into the text. The authors could better hypothesize its use by describing the usefulness of presented results.

Comments on the Quality of English Language

Authors should review their English to improve the quality of the manuscript. They should better express the concepts and describe the various steps of the analyses of the considered patient population.

Author Response

Comments 1: The authors present a retrospective study of a population with lung cancer patients. The aim of the study is to describe the characteristics determining the cancer prognosis. They considered biological and clinical factors in univariate and multivariate analyses.
The topic is of interest, however there are many criticisms:
Response 1: Thank you for recognizing the relevance of our study on prognostic factors in lung cancer patients. We appreciate the opportunity to strengthen our work through addressing the points you raised.

Comments 2: Title, line 2-3_ The title of the manuscript should be catchier and better present the purpose of the study inviting the reader to the topic.
Response 2: Thank you for your constructive feedback regarding the manuscript title. We appreciate your suggestion to make it more engaging and reflective of the study's purpose. We understand that a compelling title can enhance reader interest and better convey the focus of our research. We propose: “Assessing Prognosis: Factors Influencing Outcomes in Hospitalized Lung Cancer” (Lines 2-3).

Comments 3: Simple summary_ Line 15_ 'In our model', should be replaced with 'In our study population'
Abstract_ The sentence 'In the multivariate analysis only were independent prognostic factors admission due to tumor-related causes () and ECOG≥2.'  -Line 29-31_ should be reviewed.
Response 3: Thank you for your helpful comments regarding the simple summary and abstract. We will make changes for greater clarity (Line 15 and 29-31).

Comments 4: Introduction_ This section contains useful information to better understand the topic, however the authors could present in a less schematic way (Ref. line 45-55 and 69-74. Furthermore, the authors should indicate the meaning of the ‘*’ in Table 3.
Response 4: Thank you for your insightful comments regarding the introduction. We appreciate your suggestion to present the information in a more narrative format. To enhance readability and coherence, we will revise the section to integrate the following points into a unified discussion (Lines 43-54). We will indicate the meaning of “*” in the table (Line 139).

Comments 5: Results_ This section could be improved. For the reader the text is not easy to understand, because it is not very fluent in the terms and data presented. For this reason, they should be reorganized, also to better introduce the algorithm proposed for the management of hospitalized patients with lung cancer.
Response 5: Thank you for your valuable feedback regarding the results section. We have revised this part of the manuscript to enhance clarity and flow (Lines 167-195). We aimed to present the data in a more coherent manner, ensuring that the key findings are easily understandable for the reader. The revised section now integrates the ROC curve analyses, univariate and multivariate findings, and their significance in a more narrative format. We believe these changes will improve the overall readability of the results. We appreciate your guidance in helping us strengthen this aspect of our paper.

Comments 6: Discussion_ This section probably already revised should be further improved. Data from other studies should be better presented in parallel with the results of the retrospective analysis of the considered population. The flow algorithm presented in figure 3 should be better described into the text. The authors could better hypothesize its use by describing the usefulness of presented results.
Response 6: We have revised the discussion to better integrate findings from relevant literature alongside our results. We clearly indicate how our findings compare with those from other studies, such as the differences in hospitalization rates and mortality outcomes. This comparative analysis enhances the context and significance of our findings.
Additionally, we have expanded the description of the flow algorithm presented in Figure 3. In the revised discussion, we explain its structure and purpose more clearly, emphasizing how it can assist clinicians in managing hospitalized lung cancer patients based on identified prognostic factors. This includes detailing how the algorithm aids in risk stratification and the integration of palliative care strategies.

Reviewer 3 Report (New Reviewer)

Comments and Suggestions for Authors

Ethical Committee Aproval is not mentioned

Tumor-related causes are not defined and analyzed

The article mentioned 50% of lung cancer patients 48 have three or more comorbidities, such as chronic obstructive pulmonary disease (COPD) 49 or cardiovascular disease but no analysis about smoking +/- COPD was performed

Cause of death was lung cancer or comorbidities?

The term organ fragility has other meaning in English Medical Literature. So, it could be replaced Maybe organ dysfunction or failure ?!

Author Response

Comments 1: Ethical Committee Aproval is not mentioned
Response 1: Thank you very much for pointing this out. It is indicated at the end of the text (Line 323-325). However, in order not to facilitate this problem we will also indicate it in the methodology (Line 109-110).

Comments 2: Tumor-related causes are not defined and analyzed
Response 2: Thank you for this comment. To avoid any confusion, we have defined "tumor-related" causes of admission in our manuscript as issues arising directly from the tumor, including cachexia, atelectasis, lymphangitis, pleural effusion, organ failure due to metastasis (e.g., neurological symptoms, hepatic failure), paraneoplastic syndrome, or the need to initiate oncological treatment during hospitalization (Lines 105-108). Attempting to disaggregate these causes for sub-analysis could lead to subjectivity depending on how they are grouped/ungrouped.

Comments 3: The article mentioned 50% of lung cancer patients 48 have three or more comorbidities, such as chronic obstructive pulmonary disease (COPD) 49 or cardiovascular disease but no analysis about smoking +/- COPD was performed
Response 3: Thank you for this observation. We recognize the prognostic importance of smoking status and comorbidities. Nevertheless, patients with non-smoking lung cancer are associated with driver mutations (an aspect that we do collect) and have a different prognosis which could interfere with the results of our sample. We recognize this limitation in lines 300-301.

Comments 4: Cause of death was lung cancer or comorbidities?
Response 4: Thank you for pointing this out. It could be discussed in view of the fact that we did not indicate the cause of death of the patients. However, most patients are admitted due to tumor-related causes and have a worse prognosis. If mortality were due to comorbidity, admissions would predominate due to infections (COPD exacerbations) or heart failure (due to cardiovascular pathology).

Comments 5: The term organ fragility has other meaning in English Medical Literature. So, it could be replaced Maybe organ dysfunction or failure ?!
Response 5: Thank you for your suggestion. We agree that "organ fragility" may not convey the intended meaning in English medical literature. We have revised the term to "organ dysfunction" to improve clarity and ensure accuracy in describing the patients' condition (Line 57).

Round 2

Reviewer 1 Report (Previous Reviewer 2)

Comments and Suggestions for Authors

Most of the comments were not satisfactorily addressed, especially the last suggestion. Please re-write the article and resubmit according to the last comment.

Author Response

Comments 1: "Most of the comments were not satisfactorily addressed, especially the last suggestion. Please re-write the article and resubmit according to the last comment."

Response 1:  Thank you very much for the effort dedicated to our project. We regret that our explanations and comments were not satisfactory. In view of the fact that in the previous review we have passed the approval of the rest of the reviewers, we will try to focus on your previously commented aspects.
If there are specific aspects of our responses or manuscript that you feel remain insufficiently addressed, we would be grateful for detailed guidance so we can make the necessary improvements. We highly value your input and are committed to revising the manuscript further to meet the highest standards of clinical relevance and scientific rigor. We will highlight the new information with yellow underlining.

-----------------------------------------

Comments 1: Congratulations to you on your novel thought related to the caring of those unplanned hospital admission of lung cancer patients and the detailed analysis seeking out factors of survival and the way of improving. The following are my comments.

Response 1: Thank you very much for the effort to correct our work. Your contributions allow us to improve.

Comments 2: (1) In the text : Survival was not significantly associated with stage 4 disease. Comparing non-stage 4 with stage 4 disease, those non-stage 4 had a lower 30-day mortality(p=0.004). Do you think the two sentences above are contradictory ?

Response 2: Thank you for pointing this out. We agree with this comment. Stage IV status itself did not have a significant impact on overall survival in the multivariate analysis. However, when analyzing the stage IV vs. non-stage IV subgroups, it was seen that the proportion of 30-day mortality was higher in the stage IV subgroup. Stage IV patients constituted the majority of our population (84.8%). It may be that the size of the non-stage IV subpopulation was so small that we did not have sufficient statistical power to detect differences. We reflect these problems and the limitation they pose in the lines 262-267 of discussion.

Response 2.2: Thanks for pointing this out. We believe that we can be more explicit about the problem of the overrepresentation of stage IV in our sample (almost 85% of the total). This is a reflection of the aggressive behavior of the disease since many patients are initially diagnosed as stage IV (31) and patients with locoregional tumors subsequently progress to stage IV (32). However, this means that we may not have sufficient statistical power to detect a difference in the survival analysis.
We reflect these problems and the limitation they pose in the lines 263-271 of discussion.

  1. CDC. United States Cancer Statistics. 2024 [cited 2024 Nov 24]. U.S. Cancer Statistics Lung Cancer Stat Bite. Available from: https://www.cdc.gov/united-states-cancer-statistics/publications/lung-cancer-stat-bite.html
  2. Consonni D, Pierobon M, Gail MH, Rubagotti M, Rotunno M, Goldstein A, et al. Lung Cancer Prognosis Before and After Recurrence in a Population-Based Setting. JNCI: Journal of the National Cancer Institute. 2015 Jun 1;107(6):djv059.

Comments 3: (2) Old frail patients with metastasis, adjusting the dosage or number of chemotherapy drugs can benefit. I think this is a treatment normality.

Response 3: Thank you for your observation. We agree that dose adjustment or modification in chemotherapy regimens is a necessary practice for frail, older patients with metastatic disease, aligning with standard treatment protocols. We aim to approach hospitalization in the lung cancer patient as an opportunity to assess the need to adjust treatment. Especially if there are clinical or analytical data of risk. We have corrected some aspects of the algorithm to reflect this (Line 297).

Response 3.2: Thank you for your continued feedback and for pointing out areas that require further clarification. We fully agree that this practice is a cornerstone of personalized cancer treatment. Our intention was not to propose a deviation from standard protocols but rather to emphasize that hospitalization provides a critical opportunity for re-evaluation and optimization of ongoing treatment plans, particularly in high-risk patients. To address your concerns, we have refined our algorithm and added further clarification in the manuscript to reflect this perspective more explicitly (Lines 227-228, 292-294 and 303).

  1. Lund CM, Vistisen KK, Olsen AP, Bardal P, Schultz M, Dolin TG, et al. The effect of geriatric intervention in frail older patients receiving chemotherapy for colorectal cancer: a randomised trial (GERICO). Br J Cancer. 2021 Jun;124(12):1949–58.
  2. Hall PS, Swinson D, Cairns DA, Waters JS, Petty R, Allmark C, et al. Efficacy of Reduced-Intensity Chemotherapy With Oxaliplatin and Capecitabine on Quality of Life and Cancer Control Among Older and Frail Patients With Advanced Gastroesophageal Cancer: The GO2 Phase 3 Randomized Clinical Trial. JAMA Oncology. 2021 Jun 1;7(6):869–77.
  3. Mohamed MR, Rich DQ, Seplaki C, Lund JL, Flannery M, Culakova E, et al. Primary Treatment Modification and Treatment Tolerability Among Older Chemotherapy Recipients With Advanced Cancer. JAMA Network Open. 2024 Feb 15;7(2):e2356106.

Comments 4: (3) In my opinion, improvement of survival and quality of life comes from the regular follow-up of patients in the out patient clinic and not improving the care after the unplanned hospital admission. Follow-up with blood tests including CBC, liver and renal functions, tumor markers, body weight, CAT scan, brain MRI and bone scan in 6-month interval. Hospitalization for further examinations in case of abnormal findings. Treat after unplanned admission will be to late.

Response 4: Thank you for your valuable insight. We agree that regular, proactive follow-up is essential to improving survival and quality of life for lung cancer patients by potentially preventing urgent hospitalizations through early detection of complications. Unfortunately, as observed in our study, lung cancer patients are the oncology population with the highest volume of hospitalization. This also implies a poor prognostic situation. Hospitalization may represent a sentinel event to rethink continuous outpatient care. By identifying the factors associated with these admissions, we aim to enhance our understanding of at-risk patient profiles, which could inform more tailored follow-up strategies in the outpatient setting. This approach may ultimately contribute to optimizing long-term care and support timely intervention before urgent admission becomes necessary.

Response 4.2: Thank you very much for raising this point. We think our discussion can be enriched by this topic. Although it is true that there is no randomized scientific evidence regarding the best post-admission strategy:

  • We think that inpatient care is likely an inevitable part of the cancer journey, and optimizing supportive procedures during hospitalization should be a key focus of medical oncology (38).
  • We think, like other authors, that hospitalisation can be a sentinel event to reconsider the attitude we were having towards the oncology patient (10).
  • There are studies that show improvement in quality of life as well as a survival benefit with comanagement of oncology and palliative care physicians (29).

We adress this problem and the limitation it poses in the lines 321-325 of discussion.

  1. Numico G, Cristofano A, Mozzicafreddo A, Cursio OE, Franco P, Courthod G, et al. Hospital Admission of Cancer Patients: Avoidable Practice or Necessary Care? PLoS One. 2015 Mar 26;10(3):e0120827.
  2. Rocque GB, Barnett AE, Illig LC, Eickhoff JC, Bailey HH, Campbell TC, et al. Inpatient hospitalization of oncology patients: are we missing an opportunity for end-of-life care? J Oncol Pract. 2013 Jan;9(1):51–4.
  3. Temel Jennifer S., Greer Joseph A., Muzikansky Alona, Gallagher Emily R., Admane Sonal, Jackson Vicki A., et al. Early Palliative Care for Patients with Metastatic Non–Small-Cell Lung Cancer. New England Journal of Medicine. 2010;363(8):733–42.

Comments 5: (4) Although nutrition was mentioned, actually, body weight measuring is a very sensitive factor for monitoring of the patients' cancer status.

Response 5: Thank you for highlighting the importance of body weight monitoring as a sensitive indicator of cancer status. We acknowledge that refractory cachexia is a poor prognostic situation. The chronic inflammation that is the basis of this process is what the immunoinflammatory scores are intended to reflect. These scores are indicated in the ESMO guidelines as prognostic factors in advanced cancer (doi: 10.1016/j.esmoop.2023.101195). On the other hand, the diagnosis of some elements of cachexia (especially sarcopenia) can be complex and immunoinflammatory scores can be a simple measure to try to reflect chronic inflammation. In future considerations, we could include specific recommendations for monitoring body weight in our proposed outpatient follow-up care, as this could provide valuable insights into disease progression and help guide timely interventions.

Response 5.2: Thank you for your thoughtful comment regarding the potential utility of weight measurement in assessing patient outcomes. While weight can indeed be an important clinical parameter, we chose to focus on inflammatory scores in our study due to several key considerations.

Inflammatory scores, are objective and easily obtainable metrics, which is particularly advantageous in retrospective studies as they minimize potential biases associated with incomplete or inconsistent documentation of weight changes. Moreover, these scores have been extensively validated as prognostic factors for survival, demonstrating predictive value independent of tumor stage or primary site across various cancer types.

Importantly, these scores also reflect the chronic inflammation underlying cancer cachexia, a multifactorial syndrome that extends beyond mere weight loss to include systemic metabolic and inflammatory changes. By capturing this broader aspect of cancer-related cachexia, inflammatory scores provide a more comprehensive tool for understanding prognosis in oncology patients.

We acknowledge the value of weight measurement in certain contexts and appreciate your suggestion. However, we believe that our focus on inflammatory markers aligns with the study's retrospective design and its objectives of identifying robust, widely applicable prognostic factors. Routine clinical monitoring of systemic inflammation markers could help identify patients at risk of developing cachexia, even those without weight loss and with good performance status.

We address this problem and the limitation it poses in the lines 312-320 of discussion.

  1. McMillan DC. Systemic inflammation, nutritional status and survival in patients with cancer. Curr Opin Clin Nutr Metab Care. 2009 May;12(3):223–6.
  2. McGovern J, Dolan RD, Skipworth RJ, Laird BJ, McMillan DC. Cancer cachexia: a nutritional or a systemic inflammatory syndrome? Br J Cancer. 2022 Aug;127(3):379–82.

Comments 6: (5) I suggest the authors should implement the new strategy and compare the results with that of the previous existing one , to see if there is improvement in survival. The medical expenditure should also be contemplated for comparison.

Response 6: Thank you for your valuable suggestion. Our objective with this work is to create a rationale from which to organize a prospective project to try to improve the continuum of care of patients on admission.

Response 6.2: Thank you for pointing this out. We agree with this comment. Therefore, the next objective of our project is to apply this algorithm to our inpatient ward and see if there is an improvement in the patients' quality of life. Unfortunately, the reviewer's request exceeds the capacity of our article. The proposed comparison involves a prospective comparative study. Therefore, a new approval by the ethics committee would be necessary. It would also take a few years with the new strategy to obtain results that would allow comparison with those shown in the current article. We regret that at the present time we are unable to satisfy this condition.

This manuscript is a resubmission of an earlier submission. The following is a list of the peer review reports and author responses from that submission.

Round 1

Reviewer 1 Report

Comments and Suggestions for Authors

The paper demonstrates the outcome for a very sick group of patients with lung cancer many of them entering already in the end-of-life stage. It will be useful for readers to know what oncological or supportive treatment was used for these patients during the admission and whether those made any impact on survival or at least increased the discharge rate. For example, you mentioned that "Ninety-two (58.2%) patients received ICI". It will be interesting to know whether any tumour response was achieved. Any other comments on the effect of other treatments may be useful. There are a few patients in your cohort with an early stages. It will be interesting to know why these patients required admission contrary to the the patients with the advanced stages.

Author Response

Comments 1: [The paper demonstrates the outcome for a very sick group of patients with lung cancer many of them entering already in the end-of-life stage. It will be useful for readers to know what oncological or supportive treatment was used for these patients during the admission and whether those made any impact on survival or at least increased the discharge rate. For example, you mentioned that "Ninety-two (58.2%) patients received ICI". It will be interesting to know whether any tumour response was achieved. Any other comments on the effect of other treatments may be useful.]

Response 1: Thank you for pointing this out. We agree with this comment. Therefore, we have added information on the best response obtained with ICI treatment. In lines 134-135 we enter the following data: "The best response to ICI was complete response (n=3), partial response (n=33), stable disease (n=27) and progression disease (n=29).". This we subsequently correlated with survival in lines 168-171: "Survival was not associated stage IV. Only 24 patients of the 158 patients were non-IV stage (15.2%). When trying to find differences between stage non-IV and IV patients, stage non-IV patients had lower 30-day mortality (12.5% vs 45.5%, p=0.004) and lower mean LDH levels (306UI/L vs 421UI/L, p=0.37).".

Comments 2: [There are a few patients in your cohort with an early stages. It will be interesting to know why these patients required admission contrary to the the patients with the advanced stages.]

Response 2: Agree. We have, accordingly, added information to emphasize this point. We study this phenomenon on lines 168-171: "Survival was not associated stage IV. Only 24 patients of the 158 patients were non-IV stage (15.2%). When trying to find differences between stage non-IV and IV patients, stage non-IV patients had lower 30-day mortality (12.5% vs 45.5%, p=0.004) and lower mean LDH levels (306UI/L vs 421UI/L, p=0.37).". On lines 230-234, in the discussion, we add some arguments about this finding: "Another noteworthy aspect is that stage IV is not associated with survival. Our sample probably overestimates the stage IV population due to selection bias since stage IV patients are at higher risk of hospitalization. Nevertheless, we must bear in mind that stage IV patients may account for up to half of the cases at diagnosis and a significant proportion of locoregional cases will progress to stage IV disease (29)."

Thank you very much for your dedication and corrections.

Reviewer 2 Report

Comments and Suggestions for Authors

Congratulations to you on your detailed analysis of the factors relating patients's survival after their hospitalization. The entire manuscript was well written with acceptable language and layout.  My comments are as follow.

(1) Your conclusion is ECOG is the only independent prognostic factor for his survival. When a patient's ECOG is high that means he is in a serious condition. And as a matter of course his suvival is low. 

(2) There is no analysis how the cancer stages relating to the survival.  Why ?

(3) Some sentences need rephrase due to weird structure.

(4) I don't think your results and conclusion will give any contribution to the future clinical practice.

Author Response

Comments 1: [Congratulations to you on your detailed analysis of the factors relating patients's survival after their hospitalization. The entire manuscript was well written with acceptable language and layout.  My comments are as follow.
(1) Your conclusion is ECOG is the only independent prognostic factor for his survival. When a patient's ECOG is high that means he is in a serious condition. And as a matter of course his suvival is low.]

Response 1: Thank you for pointing this out. We agree with this comment. Importantly, in our sample of hospitalized patients. Around two thirds had an ECOG equal to or greater than 2. This is consistent with the idea that hospitalized patients present a situation of clinical deterioration that should cause us to be alarmed because it implies a poor prognosis. In this context, the ECOG seems more robust and outweighs any score or even the stage of the disease.

Comments 2: [There is no analysis how the cancer stages relating to the survival]

Response 2: Thank you for pointing this out. We focused on survival since admission and in lines 158-164 we had written that: “In univariate analysis, poorer mOSSA was associated with admission due to tu-mor-related causes -43%- (1.33 vs 7.30 months, p<0.001, HR 2.19, 95%CI 1.53-3.15), ECOG≥2 (2.43 vs 8.50 months, p<0.001, HR 2.15, 95%CI 1.42-3.25), NLR≥6 (1.87 vs 7.40 months, p=0.006, HR 1.67, 95%CI 1.16-2.42), PNI≤40 (1.67 vs 4.97 months, p=0.044, HR 1.49, 95%CI 1.01-2.20) and LDH≥210 (2.27 vs 7.87 months, p=0.044, HR 1.68, 95%CI 1.01-2.79) (Figure 2). Survival was not associated with sex (p=0.84), age (p=0.94), stage IV (p=0.88), number of treatment lines (p=0.65), CONUT (p=0.70) or mGPS (p=0.07).”

Nevertheless, the other reviewer has pointed this out to us so we probably would not have covered it correctly. We have, accordingly, added information to emphasize this point. We study this phenomenon on new lines 168-171: "Survival was not associated stage IV. Only 24 patients of the 158 patients were non-IV stage (15.2%). When trying to find differences between stage non-IV and IV patients, stage non-IV patients had lower 30-day mortality (12.5% vs 45.5%, p=0.004) and lower mean LDH levels (306UI/L vs 421UI/L, p=0.37).". On lines 230-234, in the discussion, we add some arguments about this finding: "Another noteworthy aspect is that stage IV is not associated with survival. Our sample probably overestimates the stage IV population due to selection bias since stage IV patients are at higher risk of hospitalization. Nevertheless, we must bear in mind that stage IV patients may account for up to half of the cases at diagnosis and a significant proportion of locoregional cases will progress to stage IV disease (29)."

Comments 3: [Some sentences need rephrase due to weird structure.]

Response 3: Agree. We will do a new correction. Clarity is necessary in an article. Could you point out any phrase that has drawn your attention?

Comments 4: [I don't think your results and conclusion will give any contribution to the future clinical practice]

Response 4: Unfortunately you are probably right. However, the objective of the article is to point out that patients with lung cancer who require hospitalization constitute a population with a poor prognosis. Especially if the reason for admission is related to problems derived from the tumor or the patient presents functional deterioration. This could help us adjust the patient's care or serve as a starting point to try to search for new biomarkers since, as it seems in our sample, the immunoinflammatory scores that we currently have (and that even appear in the ESMO guidelines) do not seem to improve clinical data. Furthermore, as seen in Table 5, the study of prognosis in the oncology population continues to be a topic of debate. Taking into account that lung cancer is one of the most frequent tumors and with the highest mortality, it is a topic that may be of scientific interest.

Thank you very much for your dedication and corrections.

Round 2

Reviewer 2 Report

Comments and Suggestions for Authors

Thank you for your detailed address. 

However, I have to say that your conclusion will not have any contribution to the clinical practice, unless you have to prove by processing a reformed treatment policy for lung cancer re-admission patients according to the results and philosophy of this article that its outcome is better than that of the pre-reformed one. That will be convincible.

Author Response

Authors' Responses to Reviewer's Comments (Reviewer 2)

Round 2

Comments 1: [Thank you for your detailed address.
However, I have to say that your conclusion will not have any contribution to the clinical practice, unless you have to prove by processing a reformed treatment policy for lung cancer re-admission patients according to the results and philosophy of this article that its outcome is better than that of the pre-reformed one. That will be convincible.]

Response 1: Thank you for pointing this out. We agree with this comment. It has allowed us to improve our manuscript. Our study aims to take a critical first step by identifying key prognostic factors. With this information we have developed an action algorithm for the hospitalized lung cancer patient. Our study contributes by providing the groundwork and rationale for future prospective studies, which could be designed in collaboration with clinical teams and tested in real-world settings.
